# A Clustering Baseline for Object-Centric Representations

## Abstract

Object-centric learning aims to discover and represent visual entities as a small set of object embeddings and masks, which can be later used for downstream tasks. Recent methods for object-centric learning build upon vision foundation models trained with self supervision because of the rich semantic features they produce. However, they often involve additional training to optimize for object mask accuracy for a specific granularity of objects on a test dataset, while overlooking the evaluation of the quality of the object embeddings which is arguably more important. In this work, we demonstrate how to discover objects and parts with a simple multi-scale application of k-means to the features of an off-the-shelf backbone. Our method is fast and flexible, produces interpretable masks, preserves the quality of the backbone embeddings, does not require additional training, and can capture different part/whole structures. We evaluate the quality of the obtained representation on a variety of downstream tasks including scene classification and action recognition in videos, showing that it surpasses the performance of fine-tuned object-centric learning methods. Object masks produced by our method also effectively capture real-world objects and parts at various granularity, with comparable quality to specialized methods when evaluated on unsupervised segmentation benchmarks. These results suggest rethinking the current approach to object-centric learning, with a greater focus on the quality of the representation.

## 1 Introduction

Self-supervised learning (SSL) enables computer vision models to learn from vast amounts of unlabeled data with minimal supervision. At its inception, SSL relied on handcrafted pretext tasks to learn useful representations from data, such as predicting augmentations (Zhang et al., 2016; Gidaris et al., 2018), reconstructions (Kingma & Welling, 2014), or contrastive assignments (Chen et al., 2020; Grill et al., 2020). Over time, the field has shifted towards more streamlined architectures (Dosovitskiy et al., 2020) and training objectives (Caron et al., 2021; Radford et al., 2021; He et al., 2022; Assran et al., 2023; Oquab et al., 2023), which have shown impressive performance on a wide range of downstream tasks. However, existing SSL models focus primarily on learning *global* or *local* representations, expressed respectively as a single embedding – a vector – that represents the entire image (Chen et al., 2020; Grill et al., 2020; Caron et al., 2021) or a dense feature map that represents local image patches (Zhou et al., 2022; Oquab et al., 2023). These two levels of representation are useful for a wide range of tasks, such as image classification, object detection, and segmentation. However, they lack an understanding of the structure in the represented parts, objects, and groups of entities that we typically use to describe the world (Whitehead, 1985; Spelke & Kinzler, 2007; Johnson-Laird, 2010). Bridging the gap between the two levels, *object-centric* representations aim to offer a more structured and interpretable representation of the world (Lake et al., 2017; Greff et al., 2020; Hinton, 2021; LeCun, 2022). Informally, object-centric representations are a set of features associated to distinct parts or objects in an image. Downstream tasks can benefit from such a representation in terms of compute, interpretability, and memory usage.

Learning object-centric representations remains at present an open challenge. Supervised or weakly-supervised approaches struggle with the compositional explosion of annotations. For example, SAM (Kirillov, 2023) is trained explicitly for object segmentation, but requires extensive annotations of every object in an image. Text-supervised models like CLIP (Radford et al., 2021) can be used for open-vocabulary segmentation, assuming that the training captions contained a detailed description

of every possible item. On the other hand, self-supervised approaches need to acquire a notion of *objectness* from the data itself. Such general notion is by itself not trivial, as the definition of objects can be ambiguous and task-specific (*e.g.* distinguishing between a car and a truck, or the wheel of a car and the car itself), which in turn makes it harder to define clear benchmarks and evaluation metrics. To address these challenges, early object-centric methods made use of specialized architectures (Hinton et al., 2018; Locatello et al., 2020) and training objectives (Burgess et al., 2019; Greff et al., 2019) with built-in inductive biases to better steer the model towards the desired behavior. More recent methods build upon pre-trained vision backbones – more powerful and general than ad-hoc models – and learn additional adapters to extract object-centric representations (Vo et al., 2020; Hamilton et al., 2021; Seitzer et al., 2023). Even though object-centric learning aims at producing both meaningful object-centric representation and object masks, these works have mostly been focusing on generating high-quality object masks, and evaluate them for the unsupervised segmentation task. This work aims to re-establish the two-sided nature of object-centric learning, with focus on the quality of both the object masks and the representation.

To this end, we consider a simple yet effective clustering-based method for obtaining high-quality object-centric representations. Such representations are crucial for downstream tasks that require an in-depth understanding of the objects in a scene, their position and relations. Section 3 describes our method: Given patch features extracted from *state-of-the-art* vision models such as DINOv2 (Oquab et al., 2023), we apply multiple rounds of $k$-means (Lloyd, 1982) to obtain a small set of cluster centroids that represent objects at different levels of granularity, each associated with an object mask that localizes it in the image. In Section 4, we provide a comprehensive evaluation of our approach, focusing both on unsupervised segmentation tasks and on assessing the quality of the learned representations, which is often overlooked in the object-centric literature. We show that our simple approach can rival existing object-centric methods without complex pre-training losses, lengthy fine-tuning runs, or dataset-specific hyperparameters. Our clustering-based representations yields excellent performance in downstream tasks such as image and video classification, which require compositional understanding of complex scenes and frame sequences, at a much lower computational cost than processing the dense representation. Following previous work (Locatello et al., 2020; Hénaff et al., 2022), we also evaluate how well our object masks correspond to real-world objects on common unsupervised segmentation benchmarks. We demonstrate that even though $k$-means assignment masks are not as clean as those produced by complex, specialized methods, they are more than capable of capturing objects and parts at different levels of granularity, yielding a flexible and interpretable representation of the image.

## 2 RELATED WORK

**Self-supervised learning.** The goal of self-supervised learning (SSL) is to learn useful representations from unlabeled collections of images. At a high level, this is achieved by training a model to solve a pretext task with no external annotation through which the model is encouraged to learn useful features that generalize well to other tasks. Early examples include denoising (Vincent et al., 2008), colorization (Zhang et al., 2016), or inverting geometric augmentations (Noroozi & Favaro, 2016; Gidaris et al., 2018). Modern SSL methods can be categorized into two families depending on the pretext task and the domain in which their losses are computed. Generative models are based on the auto-encoder paradigm (Vincent et al., 2008), in which the model tries to reconstruct an image from its altered versions (Kingma & Welling, 2014; He et al., 2022; El-Nouby et al., 2024). Latent-space methods avoid pixel reconstruction and formulate their loss directly in the model's embedding space. Notable examples include contrastive learning (van den Oord et al., 2018; Chen et al., 2020; Grill et al., 2020), self-distillation at the global or patch level (Caron et al., 2021; Zhou et al., 2022), or predictive methods (Assran et al., 2023). SSL features exhibit an ability to capture structure and semantics of the input (Caron et al., 2021), which can be further developed using explicit object-centric losses (Hénaff et al., 2022), additional tokens (Darcet et al., 2023), or used *as-is* for object discovery, as discussed below. This work builds on DINOv2 (Oquab et al., 2023) and seeks to efficiently extract a set-structured representation of objects and parts from the learned patch features.

**Object discovery.** Given an unlabelled set of images, object discovery aims at grouping images that contain similar objects together while also localizing these objects in forms of bounding boxes or masks. Early works focus on discovering object categories, using techniques such as probabilistic modeling (Weber et al., 2000; Sivic et al., 2005; Russell et al., 2006) or non-negative matrix fac-

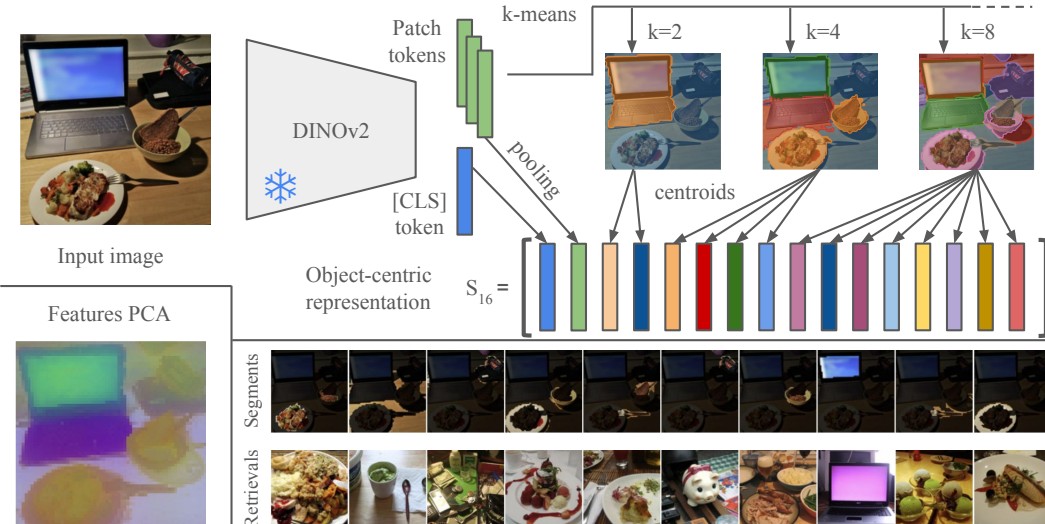

**Figure 1:** Dense features produced by a strong SSL backbone are clustered into distinct objects and parts using the classic $k$-means algorithm. **Top**: our approach: we cluster DINOv2 features using $k$-means for $k \in \{2, 4, 8\}$. The segment boundaries roughly align with object boundaries, capturing part-whole hierarchies as exemplified by the laptop, screen and keyboard segments. **Bottom left**: PCA visualization of DINOv2 features. **Bottom right**: cutouts of segments and the closest ImageNet sample retrieved using the centroid as query.

torization (Tang & Lewis, 2008), or consider only simple scenarios with a few object classes (Zhu et al., 2012; Zhang et al., 2015). Cho et al. (2015) studies object discovery in the more challenging in-the-wild image domain, proposing an algorithm that alternates between finding similar images and localizing objects. This approach is further developed into different optimization formulations (Vo et al., 2019; 2020; 2021; Choudhury et al., 2021) that could discover multiple objects per image and handle million-scale datasets. With the advent of powerful SSL backbones which already excel at at grouping similar images (Caron et al., 2021, DINO), the focus shifts entirely to the localization task. LOST (Siméoni et al., 2021) is the first to leverage DINO for finding objects with a graph-based method. Following works improve upon LOST with spectral clustering (Wang et al., 2023b; Melas-Kyriazi et al., 2022; Rambhatla et al., 2023; Wang et al., 2023a) or mask/feature correspondence distillation (Hamilton et al., 2021; Van Gansbeke et al., 2022; Zadaianchuk et al., 2022). From a methodological perspective, this work is also based on processing dense features learned by SSL backbones, though the focus is on representing objects rather than localizing them exactly. As most object discovery methods do not output object embeddings, they are not directly comparable to our work, which is instead more aligned with the object-centric learning literature.

**Object-centric learning.** This line of work evolved in parallel to object discovery and shares similar goals, though the focus is on the learned representations rather than object localization. Many object-centric methods rely on compositional generative models (Greff et al., 2019; Burgess et al., 2019) whose latent space is structured around a small set of vectors, each representing a distinct object or part of the scene. In their seminal work, Locatello et al. (2020) introduce the term *slots* to describe the attention-based set-structured bottleneck of an auto-encoder that learns to group spatial features into semantically-coherent regions. Subsequent work improve the bottleneck mechanism (Chang et al., 2022) and the decoder (Singh et al., 2022a; Wu et al., 2023; Jiang et al., 2023), with extensions to videos (Kipf et al., 2021; Singh et al., 2022b; Wu et al., 2022; Sun et al., 2023) and 3D data (Stelzner et al., 2021; Sajjadi et al., 2022). In an effort to scale from simple synthetic data to complex natural images, more recent works replace the pixel reconstruction objective with optical flow prediction (Kipf et al., 2021) or reconstruction of semantic features from pretrained SSL models (Seitzer et al., 2023; Kakogeorgiou et al., 2024). In this work, we borrow ideas and terminology from the object-centric literature, but we depart from the training-based approach, which requires specialized architectures and dataset-specific hyperparameters. We instead probe whether pre-trained SSL models already expose object-centric representations that can be extracted with a simpler, more flexible and more interpretable clustering-based algorithm.

## 3 METHOD

### 3.1 GOAL AND PROBLEM SETUP

Given a set of unlabeled images, object-centric learning aims to extract a representation that captures distinct objects or parts in a scene, possibly learned directly from the data without supervision. In line with open-ended nature of SSL methods, we use the term "object" loosely, including all parts and subparts that may be relevant depending on the context. For generality, we also allow for overlaps between the objects, to account for occlusion and part-whole object hierarchies, *e.g.* one may want to capture not only a shirt as a whole, but also its pockets and the buttons on it.

Formally, we aim to learn a function $f \colon \mathbb{R}^{H \times W \times 3} \to (\mathcal{S}_N, \mathcal{M}_N)$ that maps an image $\boldsymbol{x} \in \mathbb{R}^{H \times W \times 3}$ to a set of **object vectors** $\mathcal{S}_N = \{ \boldsymbol{s}_1, \ldots, \boldsymbol{s}_N \mid \boldsymbol{s}_n \in \mathbb{R}^D \}$ and their corresponding **assignment masks** $\mathcal{M}_N = \{ \boldsymbol{m}_1, \ldots, \boldsymbol{m}_N \mid \boldsymbol{m}_n \in \{0, 1\}^{H \times W} \}$. The role of object vectors, or *embeddings*, and masks is complementary: The former represents the "content" of the objects while the latter allows to localize them in the image. This set-structured representation complements the standard output of self-supervised models: the *global* representation $\boldsymbol{g} \in \mathbb{R}^D$, which captures the image content as a whole, and the *dense* representation $\boldsymbol{p} \in \mathbb{R}^{H \times W \times D}$, which preserves the spatial dimensions of the input by associating each coordinate to a feature vector.

The number of relevant objects in a scene depends on the data domain, the tasks at hand, as well as the users' subjective notion. For simplicity, we assume that $N$ is fixed a priori to capture all relevant objects and possibly more. In practice, this high-recall behavior fits the scenario where object tokens are fed to a downstream model, *e.g.* a classifier or a vision-language model. Processing a few more tokens than strictly necessary remains advantageous compared to the alternative of using all patch tokens. Also, hardware accelerators are better optimized to handle constant-size inputs.

### 3.2 BACKGROUND: SLOT-BASED METHODS

We briefly characterize *state-of-the-art* methods for object-centric learning such as DINOSAUR (Seitzer et al., 2023) and its follow-up SPOT (Kakogeorgiou et al., 2024). These models are implemented as an auto-encoder with a slot-attention bottleneck (Locatello et al., 2020) built on top of a pre-trained SSL backbone (Caron et al., 2021). At the bottleneck, a set of query vectors compete over spatial locations with a specialized attention mechanism to compress backbone features $\boldsymbol{p}$ into "slots" $\mathcal{S}_N$. After training, the slot vectors are used as object embeddings, and corresponding masks are obtained through an $\arg\max$ over the attention maps, associating each pixel to a single object.

Slot-based auto-encoders have proven effective in capturing objects in images, though not without limitations. **Limited flexibility:** While some formulations allow for a variable (Locatello et al., 2020) or adaptive (Fan et al., 2024) number of slots, this feature is not adopted by recent works that instead train each model with a fixed number of queries (Seitzer et al., 2023; Kakogeorgiou et al., 2024). **No-overlaps:** Predicting a single class for a given pixel can not express part-whole hierarchies, *e.g.* "wheel" and "car", which are key to understanding real world scenes (Lin et al., 2014; Zhou et al., 2017). As a consequence, a trained object-centric model is implicitly biased toward a certain granularity and object density in images, which may not generalize to other tasks and datasets. **Entangled embeddings:** Due to the reconstruction objective, slot vectors must contain both semantic and position information, which degrades downstream performance compared to the SSL backbone they are fine-tuned from (see Section 4.1). In the following section, we depart from the slot-based formulation and propose a simpler method that is able to capture a more general representation of objects and parts without training or fine-tuning.

### 3.3 CLUSTERING DENSE SSL FEATURES

Our approach stems from two observations. First, the dense feature maps of modern SSL backbones contain a rich semantic representation of the input image (Oquab et al., 2023; Darcet et al., 2023). Referencing the PCA in Figure 1, the information about objects and parts is already present, we only need to extract it. Second, the slot-based methods act as a soft and parametric approximation of $k$-means clustering (Locatello et al., 2020; Chang et al., 2022). The substantial training effort required by slot-based methods is only needed to align the learned clustering to the objects that are

annotated in the test datasets. Therefore, we investigate $k$-means clustering (Lloyd, 1982) as a simple, interpretable, flexible, and learning-free method for extracting object-centric representations.

Given the patch features produced by an SSL backbone, $k$-means groups them into $K$ clusters identified by $\mathcal{C}_K = \{c_1, \ldots, c_K\}$ centroids. Without additional processing, we directly use the centroids as object vectors and the cluster assignments and object masks. To capture objects and parts at various scales, we run $k$-means multiple times with different values of $K$, as exemplified in the top row of Figure 1. In our experiments, we build a set of object vectors that includes the global representation $g$ and all the $k$-means centroids with $K$ chosen on a geometric progression:

$$\mathcal{S}_N = \{g\} \cup \mathcal{C}_1 \cup \mathcal{C}_{2^1} \cup \ldots \cup \mathcal{C}_{2^{\log_2(N)-1}}, \tag{1}$$

where $N$ is a power of 2, and $g$ and $\mathcal{C}_1$ are associated to dummy masks that cover the whole image. The choice of $N$ and $K$ values can of course be refined if domain-specific knowledge is available.

Compared to learning-based approaches, this method based on $k$-means clustering offers several advantages. **Flexibility**: The algorithm requires no additional training or fine-tuning, and can be applied to any off-the-shelf model, allowing for quick experimentation with different backbones and number of objects. **Granularity:** Running multiple $k$-means accounts for the complexity of real-world images where each pixel can be associated to multiple objects and parts. **Semantic embeddings:** Each k-means centroid is obtained by averaging the patch features assigned to it, therefore preserving the quality of the SSL representation. Since object vectors "live" in the same space as the dense features, we can easily visualize their semantic content by retrieving related images as in Figure 1.

**Resolution and hierarchical $k$-means.** For simplicity, Section 3.1 does not distinguish between the input resolution and the resolution of the dense features. In practice, all modern SSL backbones introduce a significant downsampling, *e.g.* a DINO ViT-B/16 (Caron et al., 2021) produces a $14 \times 14$ feature map from its native resolution of $224 \times 224$. Applying $k$-means clustering at this resolution has no noticeable effect on the object vectors, but results in "blocky" object masks. Unless specified otherwise, we use the native resolution when evaluating representation quality (Section 4.1), and process the images at $4\times$ the backbone resolution for evaluating segmentation (Section 4.2). In this case, the ViT-B/16 feature map would now be $56 \times 56$, so we apply $k$-means in a hierarchical manner: an initial clustering step with $K = 256$ followed by another step at the desired $K$. Hierarchical $k$-means also has the desirable effect of reducing the bias towards equally-sized clusters, finding valid object masks even when there is an important imbalance between object sizes (Vo et al., 2024).

# 4 EXPERIMENTS

We evaluate object-centric representations along two axes: the quality of the object vectors as a compact scene representation for downstream tasks (Section 4.1), and the ability to associate each vector to an actual object in the image (Section 4.2). The former is directly related to the goal of object-centric learning, *i.e.* turning a visual input into a small set of tokens that represent a composition of entities in a scene. The latter ensures that the learned representations are indeed aligned with a human notion of "objects", and are not just a set of vectors that happens to be useful for downstream tasks. We point out that the object-centric literature often focuses on the latter, pushing to improve unsupervised segmentation metrics, while overlooking the usefulness of the learned representations. As segmentation is not the only goal, we give equal importance to both aspects in our evaluations.

## 4.1 EVALUATING OBJECT-CENTRIC REPRESENTATIONS

In the SSL literature, the standard practice to evaluate the quality of global $g \in \mathbb{R}^D$ and dense representations $p \in \mathbb{R}^{H \times W \times D}$, is to probe them with linear classifiers on a variety of tasks. This approach assumes linear separability in the feature domain and introduces a minimal number of trainable parameters, so to focus on the representation itself. We evaluate the object-centric representations in a similar spirit, but adapt the classifier to take as input an unordered set of vectors $\mathcal{S}$, which is described in Equation 1 for our approach and corresponds to slot embeddings for SPOT (Kakogeorgiou et al., 2024). Specifically, the classifier is implemented as a permutation-invariant attention pooling model:

$$y = W^{\text{out}} \left( \sum_{s \in \mathcal{S}} \frac{\exp(q^T W^{\text{key}} s)}{\sum_{s' \in \mathcal{S}} \exp(q^T W^{\text{key}} s')} W^{\text{value}} s \right), \tag{2}$$

where $q \in \mathbb{R}^D$, $W^{\{key,value\}} \in \mathbb{R}^{D \times D}$, and $W^{out} \in \mathbb{R}^{C \times D}$ are learnable parameters trained by gradient descent, and $y \in \mathbb{Y} \subset \mathbb{R}^C$ is a single- or multi-label prediction depending on the task.

### 4.1.1 IMAGE CLASSIFICATION

**Datasets and tasks.** As a first benchmark, we use ImageNet (Deng et al., 2009), the *de-facto* standard for image classification in self-supervised learning. ImageNet pictures typically contain a single predominant object and do not require compositional understanding, yet it is a useful sanity check to ensure that the object vectors preserve the quality of the SSL backbone. Moving to more complex images, we evaluate on Places205 (Zhou et al., 2014) and SUN397 (Xiao et al., 2010), two scene classification tasks that require more fine-grained understanding of objects and relationships. For these three datasets, we report top-1 accuracy. Furthermore, we construct two multi-label classification tasks from the CLEVR (Johnson et al., 2017) and COCO 2017 (Lin et al., 2014) datasets, where the goal is to predict a multi-hot vector of all object categories present in an image, *e.g.* "red large metal cube" or "wine glass". These tasks probe whether the object vectors contain sufficient information about *all* annotated objects and their attributes. For these two tasks, we report mean average precision (mAP) over 96 categories for CLEVR and 80 for COCO.

**Models.** We apply our method to DINO ViT-B/16 (Caron et al., 2021) and DINOv2 ViT-B/14 (Darcet et al., 2023), using multiple rounds of $k$-means to extract $\mathcal{S}_8$ or $\mathcal{S}_{16}$ as described in Section 3.3. For comparison, we also evaluate a linear classifier on the CLS token of these models. At the time of writing, the *state-of-the-art* model for object-centric learning is SPOT (Kakogeorgiou et al., 2024). We evaluate the open-source checkpoint based on a DINO ViT-B/16 backbone and fine-tuned on COCO using 7 slots, as well as an 8-slot and a 16-slot variant trained using the official code. Additionally, we obtain multi-scale SPOT embeddings by ensembling multiple models with 2, 4, 8 slots, the backbone CLS token as well as the average output patch. We also attempted to train SPOT with DINOv2, but it failed to converge. The second best, DINOSAUR (Seitzer et al., 2023), does not provide open-source checkpoints for evaluation.

**Results.** We observe in Table 1 that our approach largely outperforms SPOT, especially on the more object-centric tasks. Notably, all classifiers trained on slot embeddings perform worse than a linear classifier trained on the CLS token of the original backbone. We attribute this to the auto-encoding formulation of slot-based methods, which compresses both semantic and positional information into small-dimensional vectors, leading to a degradation of the backbone embeddings (Section 3.2). On the other hand, our clustering-based approach requires no additional training and improves over the CLS token performance, as the attention pooling classifiers can leverage additional structured information of each scene. In Figure 2, we assess the scaling trends for our method with respect to the number of object vectors: $\mathcal{S}_4$, $\mathcal{S}_8$, $\mathcal{S}_{16}$, or $\mathcal{S}_{32}$; and the size of the underlying DINOv2 backbone: base, large and giant. We compare against linear classifiers trained on the CLS token alone, and with attention-pooling classifiers trained on all patches (256 tokens) or on a square grid of average-pooled patches, *e.g.* $3 \times 3$ or $4 \times 4$. For simpler single-label classification tasks, performance saturates quickly with the number of objects and does not spectacularly improve over the CLS token, suggesting that a

**Table 1:** Object-centric classification: top-1 accuracy for single-label tasks, *i.e.* ImageNet, SUN 397, and Places 205, and mean average precision (mAP) for multi-label tasks, *i.e.* CLEVR and COCO 2017. The object-centric methods are grouped based on their SSL backbones, namely DINO and DINOv2, of which we report the performance of a linear classifier trained on the CLS token as a reference. The 7-slot SPOT model is the open-source release, all other SPOT models are trained by us using the official code. Our method requires no additional training, as it amounts to running multiple $k$-means clustering on the patch embeddings.

| | Tokens | ImageNet | SUN 397 | Places 205 | CLEVR | COCO 17 |
|---|---|---|---|---|---|---|
| DINO ViT-B/16 | CLS | **78.2** | **66.9** | 56.0 | 78.2 | 70.6 |
| + SPOT | 7 | 66.8 -11.4 | 59.7 -7.2 | 51.1 -4.8 | 70.8 -7.4 | 67.6 -3.0 |
| + SPOT | 8 | 67.6 -10.6 | 59.6 -7.3 | 50.8 -5.2 | 73.7 -4.5 | 67.7 -2.9 |
| + SPOT | 16 | 69.2 -9.0 | 60.3 -6.6 | 51.5 -4.5 | 81.5 +3.3 | 67.7 -2.9 |
| + SPOT ensemble | CLS+1+2+4 | 71.4 -6.8 | 59.4 -7.5 | 54.6 -1.4 | 75.9 -2.3 | 69.4 -1.2 |
| + SPOT ensemble | CLS+1+2+4+8 | 72.7 -5.5 | 59.2 -7.7 | 55.5 -0.5 | 84.4 +6.2 | 70.8 +0.2 |
| + K-means | CLS+1+2+4 | 78.1 -0.1 | 66.0 -0.9 | 58.8 +2.8 | 85.6 +7.4 | 72.3 +1.7 |
| + K-means | CLS+1+2+4+8 | **78.2** +0.0 | 66.0 -0.9 | **59.2** +3.2 | **89.7** +11.5 | **72.6** +2.0 |
| DINOv2 ViT-B/14 reg | CLS | 83.9 | 77.4 | 64.4 | 74.6 | 77.0 |
| + K-means | CLS+1+2+4 | 84.3 +0.4 | **77.9** +0.5 | 65.9 +1.5 | 84.1 +9.5 | 81.3 +4.3 |
| + K-means | CLS+1+2+4+8 | **84.6** +0.7 | 77.7 +0.3 | **66.1** +1.7 | **88.2** +13.6 | **82.3** +5.3 |

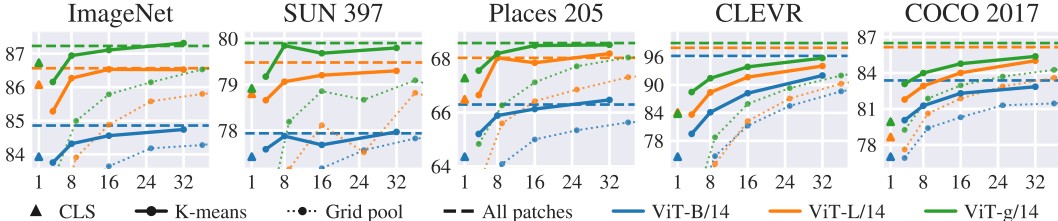

**Figure 2:** Scaling trends w.r.t. the number of object tokens and backbone size. Using open-source DINOv2 models of different sizes, we compare the classification performance when training on: the CLS token, all $16 \times 16$ patch tokens, a square grid of average-pooled patches *e.g.* $3 \times 3$, or an increasing number of object tokens extracted by running multiple $k$-means on the dense features. The object-centric representation strikes a gradual trade-off between the lower performance of the CLS token and the top-line performance of all patches. At equivalent token counts, our clustering-based representation always outperforms the pooling-based one, highlighting the importance of capturing objects/parts over simply increasing the number of input vectors.

global representation is mostly sufficient. In contrary, for more complex scenes, our approach leads to sizable gains, efficiently bridging the gap between the global and dense top-line representation. Furthermore, the comparison with average-pooled patches highlights the importance of capturing objects/parts over simply increasing the number of input vectors. Unsurprisingly, our method scales well with the size of the backbone: Since object vectors are averaged from the patch embeddings, an improvement in the backbone will directly benefit the object-centric representation.

### 4.1.2 VIDEO ACTION RECOGNITION

Object-centric representations are particularly valuable in scenarios where large spatial and temporal resolutions are crucial to performance but processing a patch-based representation would be computationally expensive. Focusing on videos, we evaluate whether a small set of vectors corresponding to objects/parts can offer a compact and expressive representation for action recognition.

**Datasets and features.** We consider two datasets for action recognition: Kinetics-400 (Carreira & Zisserman, 2018, K400) and Something-Something V2 (Goyal et al., 2017, SSv2). For each equally-spaced frame of a $T \times H \times W$ video, we apply a DINOv2 ViT-L/14 backbone (Darcet et al., 2023) and extract either: a) the CLS token alone, resulting in $T$ tokens; b) our $\mathcal{S}_N$ representation, resulting in $TN$ tokens; c) all $THW/14^2$ patches. We then add temporal embeddings to identify each frame and train a two-blocks transformer classifier on the token sequence. To demonstrate the efficiency and flexibility of our method, we sweep over several resolutions: temporal with $T \in \{8, 16\}$ frames, spatial with $H, W \in \{224, 448\}$ pixels, and "object" with $N \in \{8, 16\}$. With this range of parameters, a video can be represented with as few as 8 tokens, up to 16384, with obvious effects on performance.

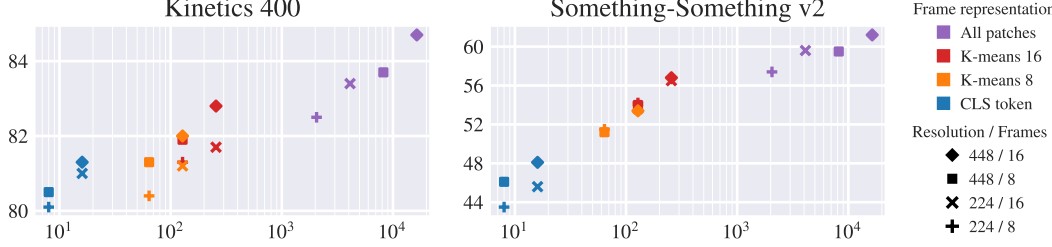

**Figure 3:** Kinetics 400 and Something-Something v2 classification accuracy *vs.* number of tokens used to represent the input (**log scale**). We consider four options to represent each frame: using only the CLS token, using all tokens (maximal performance, expensive), or using our $\mathcal{S}_8$ or $\mathcal{S}_{16}$ representation. We experiment with $T \in \{8, 16\}$ frames, and $\{224^2, 448^2\}$ resolution. Our $\mathcal{S}_N$ representation recovers most of the performance with orders of magnitude fewer tokens than the most expensive regimes, especially for Something-Something v2 where recognizing relevant objects is critical to the task. See Section 4.1.2 for more details.

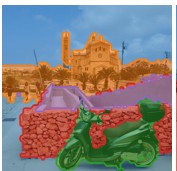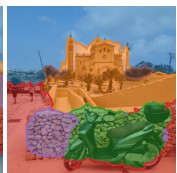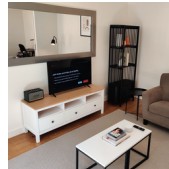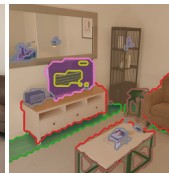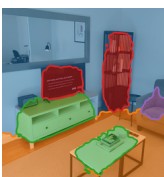

**Figure 4:** Qualitative comparison on two images between the masks obtained with k-means (middle) and SPOT, a specialized object-centric method (right). SPOT mask don't overlap and have cleaner object boundaries. K-means mask have noisier boundaries, but capture objects at different granularity and overlapping parts.

**Results.** Figure 3 reports the classification accuracy versus the total number of input tokens. The accuracy increases almost monotonically as more tokens are used to represent the video. The best performance is obtained by using all patches from $T = 16$ frames at $448 \times 448$ resolution, *i.e.* the most computationally expensive option. However, the object-centric representation recovers most of the performance with as few as 256 tokens ($T = 16$, $N = 16$), yielding 82.8% on K400 and 56.5% on SSv2, close to the top-line performance of 84.7% and 61.2%, respectively. Another benefit of the set-based representation is that increasing the spatial resolution from $224^2$ to $448^2$, boosts K400 performance with no additional compute in the classifier.

## 4.2 EVALUATING MASK QUALITY

Section 4.1 evaluates the quality of set-structured representations $\mathcal{S}_N$ for downstream tasks. However, the set structure does not imply that the representation is necessarily *object-centric*, *i.e.* aligned with a human notion of objects and parts. This defining property is the focus of this section. In previous works (Locatello et al., 2020; Greff et al., 2022; Hénaff et al., 2022; Seitzer et al., 2023; Kakogeorgiou et al., 2024), it is evaluated as unsupervised segmentation: Object masks $\mathcal{M}_N$ are compared to ground-truth object segments in annotated datasets, irrespective of the associated embedding. Though the task is framed as segmentation, the goal is not to achieve pixel-perfect masks, but rather to show that object vectors are associated to meaningful entities in an interpretable way, as in Figure 4.

**Datasets.** As in previous works, we consider a variety of datasets with different levels of complexity and annotation quality. The PASCAL VOC 2012 (Everingham et al., 2010), COCO 2017 (Lin et al., 2014), and ADE20K (Zhou et al., 2017) datasets are standard benchmarks for real-world object segmentation, with annotations for both instance-level and category-level segmentation. ClevrTex (Karazija et al., 2021) is a synthetic dataset of textured 3D shapes. MOVI (Greff et al., 2022) is a video dataset in which rendered 3D shapes move around in a textured environment. Following standard practice, each frame is treated as an independent sample and results are reported for the MOVI-C and MOVI-E variants, which differ in the complexity and number of objects in a scene.

**Metrics.** Given a set of ground-truth annotations $\{m_\tau\}_{\tau=1}^{\mathcal{T}}$ and a set of predicted masks $\{\hat{m}_n\}_{n=1}^{N}$, we first compute the intersection over union (IoU) between all $(\tau, n)$ pairs. The **mean best overlap (mBO)** is the mean IoU between each ground-truth mask and its best-matching predicted mask: $\text{mBO} = \sum_{\tau=1}^{\mathcal{T}} \max_n \text{IoU}(\hat{m}_\tau, m_n)/\mathcal{T}$. Another relevant metric is the **detection rate (DetRate)**, which considers a ground-truth object as "detected" if there exist a predicted mask with an IoU above a certain threshold, typically 0.5: $\text{DetRate} = \sum_{\tau=1}^{\mathcal{T}} \mathbb{I}\left[\max_n \text{IoU}(\hat{m}_\tau, m_n) > 0.5\right]/\mathcal{T}$. Compared to mBO, detection rate does not focus on pixel-perfect segmentation, rather it ensures that all annotated objects are appropriately identified so that they can be used for the task at hand. For datasets with

**Table 2:** Unsupervised segmentation on following Hénaff et al. (2022): mean best overlap (mBO) and detection rate (DRate). To capture objects at different scales and granularity, our method produces 255 masks by running multiple k-means with $K \in \{1, 2, 4, \dots, 128\}$ on a pre-trained DINOv1 ViT-B/14 and DINOv2 ViT-B/16. ODIN applies k-means similarly on a ViT-B/14 backbone trained from scratch with an object-centric loss. For SPOT, we obtain 255 masks from an ensemble of models based on DINOv1 ViT-B/14 and fine-tuned on COCO.

| | COCO 2017 | | | VOC 2012 | | | ClevrTex | | ADE20K | | MOVI C | | MOVI E | |
| --- | --- | --- | --- | --- | --- | --- | --- | --- | --- | --- | --- | --- | --- | --- |
| | $\text{mBO}^i$ | $\text{mBO}^c$ | $\text{DRate}^i$ | $\text{mBO}^i$ | $\text{mBO}^c$ | $\text{DRate}^i$ | $\text{mBO}^i$ | $\text{DRate}^i$ | $\text{mBO}^i$ | $\text{DRate}^i$ | $\text{mBO}^i$ | $\text{DRate}^i$ | $\text{mBO}^i$ | $\text{DRate}^i$ |
| ODIN (ViT-B/14) | 49.7 | 54.7 | 48.2 | – | – | – | – | – | – | – | – | – | – | – |
| SPOT (DINOv1 ViT-B/14) | 46.3 | 51.9 | 46.0 | 56.3 | 59.5 | 64.8 | 59.9 | 77.1 | 51.7 | 52.7 | 52.9 | 59.7 | 45.4 | 45.6 |
| K-means (DINOv1 ViT-B/14) | 50.2 | 58.9 | 51.1 | 61.3 | 65.3 | 69.8 | **74.9** | **92.0** | **58.7** | 64.4 | **66.4** | 80.3 | **57.3** | 64.6 |
| K-means (DINOv2 ViT-B/16) | **51.3** | **61.8** | **55.0** | **61.8** | **66.3** | **71.0** | 70.2 | 89.4 | 57.7 | **66.1** | 64.5 | **81.1** | 56.9 | **68.6** |

both instance-level and category-level annotations, we mark the metrics with a superscript $i$ or $c$ to distinguish which ground-truth masks they refer to.

**Evaluation protocols.** At a high level, both mBO and DetRate are recall-based metrics, *i.e.* they measure the ability of a method to *retrieve* annotated ground-truth masks. For their nature, these metrics are easily "cheated" by predicting more and more masks. Related works in the object-centric literature typically follow one of two evaluation protocols. **Total partitioning:** Locatello et al. (2020) and follow-up works constrain each pixel to belong to one and only one object, *e.g.* by taking an $\arg\max$ of the slot attention weights, effectively producing a total partitioning of the image. Under this constraint, any overlap in the ground-truth masks can not be predicted, and therefore those annotations are excluded from the metrics. Though appropriate for the synthetic images (Johnson et al., 2017; Kabra et al., 2019) considered in early methods, we argue that this evaluation protocol is not aligned with the overarching goal of object-centric learning. Scoring a high mBO merely indicates that the predictions match the annotation granularity of a dataset (*e.g.* bike *vs.* truck, wheel *vs.* car) and that the number of predictions corresponds to the average number of annotations per image. However, a model optimized for a specific dataset will hardly generalize outside its training domain. **Recall@N:** the protocol of Hénaff et al. (2022) caps the number of predicted masks to a large number, without enforcing a constraint on their spatial arrangement. This choice is better suited to real-world scenes, in which three-dimensional objects can partially occlude each other, *e.g.* the books on the table in the qualitative example of Figure 4. Furthermore, it allows to predict additional masks for parts and sub-parts that may not be annotated in a test dataset, but are nonetheless relevant. Overall, we find this second protocol to be more in line with the goal of object-centric learning: obtaining representations that generalize well to a wide range of downstream tasks.

**Results.** For ease of comparison, we evaluate our method under both protocols. To match the evaluation of Hénaff et al. (2022), we apply $k$-means multiple times with $K \in \{1, 2, 4, 8, \ldots, 128\}$ to extract a total of 255 masks per image. To compare with SPOT, we train eight SPOT models on COCO with a number of slots in $\{2, 4, 8, \ldots, 128\}$, and combine their independent predictions plus a dummy mask. Note that it is highly impractical to train and run inference on multiple SPOT models in parallel, nonetheless we provide the result for completeness. The results in Table 2 show that our simple method significantly outperforms the more specialized baselines, especially for category-level segmentation. On the other hand, Table 3 follows the evaluation protocol of Locatello et al. (2020), where each method predicts a small number of non-overlapping segments, and any overlap in the ground-truth masks is ignored. We compare with the *state-of-the-art* methods LSD (Jiang et al., 2023), DINOSAUR (Seitzer et al., 2023), and SPOT (Kakogeorgiou et al., 2024), which fine-tune a separate DINO ViT-B/16 with a fixed ad-hoc number of slots for each dataset. Instead, we run a single $k$-means on a frozen DINOv2 ViT-S/14 backbone and choose $K$ to match the number of slots of the specialized methods on each dataset. Unsurprisingly, our method falls short of the *state-of-the-art* because of the lack of dataset-specific tuning and the noisier masks produced by $k$-means. The main takeaway is that our method is more suited for a high-recall no-tuning scenario where running multiple $k$-means with different $K$ values can capture objects/parts of different sizes and granularity.

## 4.3 ABLATION OF IMAGE ENCODER

The approach presented in this work is agnostic to the image representations used. We based our analysis on DINOv2 but, in principle, similar object-centric representations could be obtained from other pre-trained backbones. The ablation study in Table 4 compares classification and unsupervised

**Table 3:** Unsupervised segmentation following the protocol of Locatello et al. (2020): any overlap in the ground-truth masks is ignored for computing mean best overlap (mBO). The baseline methods fine-tune a separate model for each dataset with a number of slots fixed ad-hoc (in parentheses). Our method uses a frozen DINOv2 backbone, and run a single $k$-means with a matching number of clusters. As confirmed by the results, this setup is suboptimal for our method, which has no dataset-specific tuning and produces noisier masks.

|  | COCO 2017 (7) | | VOC 2012 (6) | | MOVI-C (11) | MOVI-E (24) |
|---|---|---|---|---|---|---|
|  | mBO$^i$ | mBO$^c$ | mBO$^i$ | mBO$^c$ | mBO$^i$ | mBO$^i$ |
| LSD | 30.4 | – | – | – | 45.6 | 39.9 |
| DINOSAUR | 32.3 | 38.8 | 44.0 | 51.2 | 42.4 | – |
| SPOT | **35.0** | **44.7** | **48.3** | **55.6** | **47.3** | **40.1** |
| K-means | 30.4 | 38.8 | 44.0 | 50.1 | 41.8 | 36.0 |

**Table 4:** Ablation of the image encoder *w.r.t.* model family and size. For each backbone, we apply the same $k$-means clustering procedure and evaluate both classification and unsupervised segmentation tasks. Classification performance tends to increase with model size, while the trend for segmentation favors smaller models.

| Backbone | | Classification (Acc, mAP) | | Segmentation (mBO$^c$, mBO$^i$) | |
|---|---|---|---|---|---|
| Family | Size | SUN 397 | COCO 2017 | COCO 2017 | MOVI-C |
| MAE | ViT-B/16 | 60.8 | 69.6 | 42.7 | 48.1 |
| DINO | ViT-B/16 | 66.0 | 72.6 | 58.9 | **66.4** |
| DINOv2 | ViT-B/14 | **77.7** | **82.3** | **61.8** | 64.5 |
| MAE | ViT-L/16 | 62.4 | 73.1 | 56.6 | **64.7** |
| CLIP | ViT-L/14 | **80.0** | 82.0 | 51.3 | 56.4 |
| DINOv2 | ViT-L/14 | 79.2 | **84.0** | **60.1** | 63.6 |
| MAE | ViT-H/14 | 64.5 | 74.3 | 57.5 | **66.1** |
| AM-RADIO | ViT-H/16 | **80.9** | 83.8 | **61.9** | 63.5 |
| DINOv2 | ViT-g/14 | 79.7 | **84.8** | 59.1 | 62.7 |

segmentation performance of several models from other "families" including MAE (He et al., 2022), CLIP (Radford et al., 2021), DINO (Caron et al., 2021) and AM-RADIO (Ranzinger et al., 2024). Among base models, DINOv2 is a clear winner for its performance on SUN 397, COCO multi-label classification and COCO category-level unsupervised segmentation. Moving to the large models, we observe that classification performance increases consistently with model size, while the trend for segmentation is less clear. CLIP shows strong classification performance, but its dense features are noisy and do not cluster well into object-centric segments. Surprisingly, among DINOv2 models, the smallest size works best for segmentation, likely due to clustering issues when the embeddings grow larger or due to pre-training artifacts. Therefore, for the large-size class, we retain the DINOv2 ViT-L/14 because of its strong performance on both task types compared to CLIP and MAE. At the largest model size, AM-RADIO is the best performer, though it can not be compared directly to the other models as it is distilled from DINOv2, CLIP, and SAM.

## 5 CONCLUSION

We presented a clustering-based method for extracting object-centric representations of images using any pre-trained SSL backbone. The strength of the approach is in its simplicity, speed and flexibility: by applying $k$-means multiple times with different number of clusters, we are able to assemble a rich yet compact set-structured representation of objects. Our findings show that the clustering approach outperforms *state-of-the-art* object-centric learning methods in terms of representation quality and is a strong competitor in terms of unsupervised segmentation, though its masks are not pixel-perfect.

**Previous work.** Compared to the existing literature, our approach improves on several aspects: a) it does not require lengthy re-training runs to swap the backbone or adjust the desired number of objects, allowing for fast experimentation, b) it avoids learnable projections and preserves the quality of the backbone embeddings, c) object semantics are learned during the pre-training of the backbone on large-scale datasets, which avoids narrow domain biases, and d) at inference time, the only computation overhead is the clustering algorithm, which is negligible *w.r.t.* the backbone. With this work, we also aim to refocus the research on object-centric *learning* to the evaluation of its representations. The results in Section 4.1 shall serve as a baseline for more extensive and comprehensive benchmarks.

**Future directions.** The outcome of this work stands on the figurative shoulders of powerful SSL backbones. Any improvement in SSL pre-training that yields smoother, richer, and more expressive dense features will directly improve the clustering the resulting object-centric representations. As more powerful SSL backbones are developed, discovering and representing objects from their dense features using simple clustering methods may become the dominant approach. This work chooses $k$-means for its simplicity and similarity to slot attention (Locatello et al., 2020), but other methods, *e.g.* agglomerative clustering (Sibson, 1973) or DBSCAN (Ester et al., 1996), may be applicable too. Last, we focused our evaluations on scene and video classification tasks, but the potential of object-centric representations is broader. A small set of tokens that represent objects can alleviate the computational cost of training vision-language models, and serve as a compact representation for world models in reinforcement learning. This calls for standardized models and extended benchmarks, so that the community can iterate forward.

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

# A  ADDITIONAL EXPERIMENTS

## A.1  ABLATION STUDY: WHICH FEATURES TO CLUSTER FOR UNSUPERVISED SEGMENTATION?

For all experiments in the main text, we use the output features of each model as the input to the clustering algorithm. In the case of DINOv1 and DINOv2, these are the final patch features, after the last transformer block (self attention and MLP), and after the final LayerNorm operation. Other works in the unsupervised segmentation literature have proposed to use the queries, keys, and values of the last attention block instead (Siméoni et al., 2021; Wang et al., 2022; 2023b;a). In Table 5, we compare the performance of clustering-based unsupervised segmentation when using the queries, keys, and values of the last attention block, or the final patch features at the output of the model. These results extend the ones presented in Table 2 in Section 4.2. In the setting considered in this work, the patch features prove to be the best choice for clustering-based unsupervised segmentation, which we attribute to the clustering-like effect of the IBOT loss applied to patch features during DINOv2 pre-training (Oquab et al., 2023).

**Table 5:** Ablation study: which DINOv2 ViT-B/16 (Darcet et al., 2023) features work best for clustering-based unsupervised segmentation? We compare taking the queries, keys, and values of the last attention block, or taking the final patch features at the output of the model. We follow the evaluation protocol of Hénaff et al. (2022), namely clustering the spatial features multiple times with $k$-means using $k \in \{1, 2, 4, \ldots, 128\}$ to obtain 255 overlapping masks, and reporting mean best overlap (mBO) and detection rate (DRate).

| | COCO 2017 | | | VOC 2012 | | | ClevrTex | | ADE20K | | MOVI C | | MOVI E | |
|---|---|---|---|---|---|---|---|---|---|---|---|---|---|---|
| | $mBO^i$ | $mBO^c$ | $DRate^i$ | $mBO^i$ | $mBO^c$ | $DRate^i$ | $mBO^i$ | $DRate^i$ | $mBO^i$ | $DRate^i$ | $mBO^i$ | $DRate^i$ | $mBO^i$ | $DRate^i$ |
| Query | 33.8 | 39.1 | 22.7 | 43.7 | 46.5 | 35.6 | 52.5 | 53.5 | 44.8 | 38.7 | 50.2 | 47.5 | 41.0 | 31.3 |
| Key | 30.7 | 36.0 | 19.9 | 41.8 | 45.1 | 34.5 | 50.8 | 50.8 | 42.2 | 36.1 | 48.7 | 45.1 | 38.7 | 28.4 |
| Value | 26.5 | 32.1 | 16.2 | 37.6 | 41.8 | 29.4 | 39.4 | 33.5 | 35.9 | 27.6 | 38.8 | 28.9 | 27.5 | 13.5 |
| Output | **51.3** | **61.8** | **55.0** | **61.8** | **66.3** | **71.0** | **70.2** | **89.4** | **57.7** | **66.1** | **64.5** | **81.1** | **56.9** | **68.6** |

## A.2  ABLATION STUDY: VIDEO ACTION RECOGNITION ON A BUDGET

In Section 4.1.2, we demonstrate the effectiveness of using k-means centroids to represent video frames for action recognition. The hypothesis is that k-means clustering is able to isolate and represent objects, offering a compact and structured representation of each frame. We demonstrate that given a small budget of 8 or 16 tokens per frame, the k-means centroids are able to approximate the performance of a more expensive model that takes all patches of each frame as input. Given this result, one may wonder if other choices of representation can yield the same performance, while satisfying the budget constraint. In other words, we wish to investigate whether it is only a matter of token count or whether the content of these tokens is also important. We set up an additional experiment using videos at $224 \times 224$ pixel resolution, where we consider:

- dropping patches at random to fit into a given budget, *e.g.* keeping 16 patches out of 256;
- average-pooling the spatial features to fit into a given budget, *e.g.* pooling $16 \times 16$ patches to $4 \times 4$ with bicubic interpolation.

In Figure 5, we compare the performance of these different choices of representation for video action recognition on Kinetics-400 and Something-Something-V2. For both datasets and all model sizes considered, ViT-S/16, ViT-B/16, ViT-L/16, we observe that k-means centroids consistently yield better performance than other choices of representation at a given budget. We conclude that the object-centric representation provided by k-means centroids is beneficial for video action recognition, and that the performance gains can not be solely attributed to the number of tokens used.

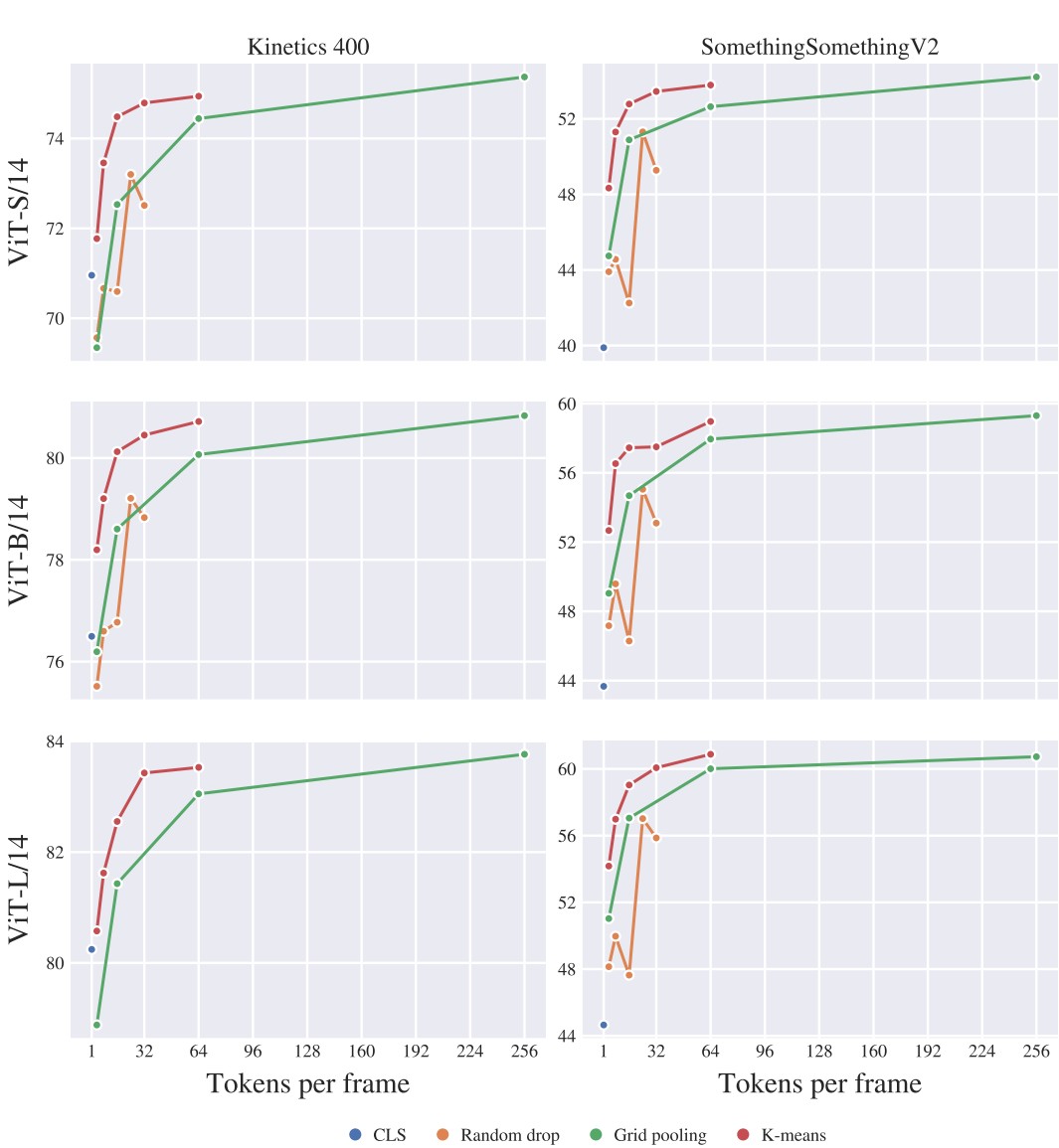

**Figure 5:** Ablation study: video action recognition on Kinetics-400 and Something-Something-V2 with a budget constraint on the number of tokens. Starting from 8 frames per video at $224{\times}224$ resolution, we compare the performance of taking k-means centroids as object representations, dropping patches at random, or average-pooling the spatial features on a square grid, *e.g.* $1{\times}1$, $2{\times}2$, $3{\times}3$, ..., $16{\times}16$. As a baseline, we also include the performance of a model that represents each frame only with the CLS token.

## A.3 TRAINING SPOT WITH A DINOv2 BACKBONE

For the experiments in Section 4, we train several SPOT models using the author's code (Kakogeorgiou et al., 2024). To confirm that our reproduction matches the official results, we train a DINOv1 ViT-B/16 model on COCO with 7 slots, as done in the original work. The unsupervised segmentation results on COCO reported in Table 6 are consistent with the official results.

**Table 6:** Unsupervised segmentation results on COCO for a SPOT model with 7 slots fine-tuned on COCO using a DINOv1 ViT-B/16 backbone.

|  | FG-ARI | $\text{mBO}^c$ | $\text{mBO}^i$ |
|---|---|---|---|
| Official | **37.8** | 44.7 | **35.0** |
| Reproduction | 36.2 | **45.0** | **35.0** |

We then attempted to train a SPOT model with more recent DINOv2 ViT-B/16 backbones of different sizes. For all models, we use COCO as the training set, but train different models with 7, 14, and 28 slots to be tested on COCO, VOC 2012, MOVI C, and MOVI E. Each model takes around 24 hours to train, both stage 1 and stage 2, on a single GPU. In this case, however, we observe a significant performance drop when using DINOv2 as the backbone, therefore we do not use these models for the experiments in the main text.

**Table 7:** SPOT models with 7, 14, and 28 slots based on either a DINOv1 or DINOv2 backbone. All models are fine-tuned on COCO and evaluated on the other datasets.

|  | COCO 2017 (7) | | | VOC 2012 (6) | | | MOVI-C (11) | | MOVI-E (24) | |
|---|---|---|---|---|---|---|---|---|---|---|
|  | FG-ARI | $\text{mBO}^c$ | $\text{mBO}^i$ | FG-ARI | $\text{mBO}^c$ | $\text{mBO}^i$ | FG-ARI | $\text{mBO}^i$ | FG-ARI | $\text{mBO}^i$ |
| DINOv1 ViT-B/16 | 36.2 | **45.0** | **35.0** | 22.0 | **58.4** | **50.3** | **64.2** | **38.8** | **60.6** | **29.5** |
| DINOv2 ViT-S/14 | 41.6 | 35.1 | 29.3 | 27.6 | 43.8 | 41.0 | 63.3 | 26.1 | 60.1 | 21.4 |
| DINOv2 ViT-B/14 | **45.1** | 33.8 | 28.9 | 36.6 | 44.2 | 41.8 | 59.8 | 22.4 | 53.9 | 18.3 |
| DINOv2 ViT-L/14 | 39.6 | 32.7 | 26.6 | **39.7** | 44.2 | 38.8 | 56.9 | 18.9 | 56.0 | 16.5 |

