# OpenReview forum: "A Clustering Baseline for Object-Centric Representations"
_ICLR.cc/2025/Conference — Submitted to ICLR 2025_

### Official Review · Reviewer_srbe · 2024-10-23

**Soundness:** 2
**Presentation:** 3
**Contribution:** 1
**Rating:** 3
**Confidence:** 5

**Summary:**

This paper introduces an object-centric framework for extracting object representations without the need for training. The proposed approach involves applying simple k-means clustering to features from off-the-shelf backbones, specifically DINOv2 [1]. The centroids from clustering, computed with different numbers of clusters (k values), are stored to capture varying levels of granularity. The resulting set of centroids is then treated as the object-centric representation of the image. These representations are evaluated on downstream tasks such as classification and video action recognition to assess their quality, and on image segmentation to determine how well they visually align with objects.

---
[1] Maxime Oquab et al., DINOv2: Learning Robust Visual Features without Supervision, arXiv

**Strengths:**

* Learning object-centric representations in an unsupervised manner is both crucial and challenging, making the task addressed in this paper highly significant.
* Evaluating object-centric representations beyond object segmentation, on downstream tasks like video action recognition, is an important demonstration of their usability. The evaluation uses diverse and comprehensive datasets.
* The paper is well-written and easy to follow.

**Weaknesses:**

* Applying clustering to pretrained encoder features is a well-known concept. Previous work in object-centric learning, such as DINOSAUR [1] and SOLV [2], has reported k-means clustering results in their ablations. Furthermore, clustering DINO [3] features has been used for segmenting objects in videos [4] and images [5, 6, 7]. This paper currently provides an evaluation of this well-established clustering approach on different benchmarks, but it is unclear how the framework represents a significant improvement or introduces novelty.

* In section 3.2, assuming overlapping masks across different granularities is an advantage over current object-centric methods, it’s worth noting that SAM [8] can extract overlapping masks for any object at any scale with pixel-level precision. This approach offers higher precision than patch clustering and can be used to extract object-centric representations in a zero-shot manner. What is the advantage of the proposed model compared to SAM in this regard?

* Since the CLS token is included in the $S_N$ representation set, it is expected to perform similarly in single-object classification tasks (e.g., ImageNet). For multi-object datasets like COCO, centroids should enhance performance by adding more granularity, whereas the CLS token summarizes the entire scene. Therefore, the comparison with the CLS token in Table 1 doesn’t provide much insight. Moreover, the approach uses multiple levels of granularity ($k = \\{1, 2, 4, 8\\}$), whereas SPOT [9] is limited to a single scale (7, 8, or 16), placing SPOT at a disadvantage. For a fair comparison, SPOT should include multi-scale representations, similar to the evaluations in Tables 2 and 3.

* For video action recognition (Figure 3), how does dropping some portion of tokens in the “all-patches” setting affect the tradeoff between token count and performance? Redundant tokens are common in video understanding tasks [10]. This comparison is important to highlight the efficiency of clustering centroids.

* The proposed approach leverages DINOv2 [11], which has been shown to significantly improve performance in object-centric tasks [2], while other models in Table 3 rely on the original DINO. Despite this advantage, both DINOSAUR and SPOT outperform the proposed method when using a single scale, i.e., a fixed number of slots/centroids, suggesting that the performance gain may primarily stem from the use of multi-scale representations. Given this, a multi-scale version of prior object-centric representations, such as a collection of DINOSAUR slots with $S = \\{1, 2, 4, 8\\}$, is expected to provide richer information than k-means clustering.

---
[1] Maximilian Seitzer et al., Bridging the Gap to Real-World Object-Centric Learning, ICLR 2023

[2] Görkay Aydemir et al., Self-supervised Object-centric Learning for Videos, NeurIPS 2023

[3] Mathilde Caron et al., Emerging Properties in Self-Supervised Vision Transformers, ICCV 2021

[4] Georgy Ponimatkin et al., A Simple and Powerful Global Optimization for Unsupervised Video Object Segmentation, WACV 2023

[5] Paul Engstler et al., Understanding Self-Supervised Features for Learning Unsupervised Instance Segmentation, arXiv

[6] Gyungin Shin et al., Unsupervised Salient Object Detection with Spectral Cluster Voting, CVPRW 2022

[7] Xudong Wang et al., Cut and Learn for Unsupervised Object Detection and Instance Segmentation, CVPR 2023

[8] Alexander Kirillov et al., Segment Anything, ICCV 2023

[9] Ioannis Kakogeorgiou et al., SPOT: Self-Training with Patch-Order Permutation for Object-Centric Learning with Autoregressive, CVPR 2024

[10] Tengda Han et al., Turbo Training with Token Dropout, BMVC2022

[11] Maxime Oquab et al., DINOv2: Learning Robust Visual Features without Supervision, arXiv

**Questions:**

Please see weaknesses.

---

> ### Author Response · Authors · 2024-11-28
> **Response to reviewer srbe**
>
> Dear reviewer `srbe`, we are grateful for your thorough review of our manuscript and appreciate the time you took to provide detailed feedback. Below, we address your main concerns and look forward to engaging in a productive discussion.
>
> **Clustering is well-known in object-centric learning. It is unclear how the framework represents a significant improvement or introduces novelty.**
>
> We thank the reviewer for the additional references that we missed, and that we will be happy to include in the paper. We agree that k-means is often included as a baseline in other works, where it is shown to yield poor results, especially if little effort is applied to optimizing it. However, our paper demonstrates that k-means is a quite valid and robust approach: it is reasonably performant for unsupervised segmentation under the protocol used in ODIN, and, more importantly, it is much stronger at representation learning, which the object-centric literature rarely evaluates.
>
> Regarding our work’s novelty and contributions, we would like to stress that our focus is on re-establishing the two-fold goal of object-centric learning: object localization and object-based representation. Other works only pay attention to the first aspect and ignore the representation learning aspect, which is arguably more important. The fact that a simple modification to k-means leads to better performance than more involved state-of-the-art baselines in representation learning suggests rethinking the current approach to object-centric learning. For a more in-depth discussion, we refer the reviewer to our general response to all reviewers above.
>
> **SAM can extract overlapping masks for any object at any scale with pixel-level precision. This approach offers higher precision than patch clustering and can be used to extract object-centric representations in a zero-shot manner. What is the advantage of the proposed model compared to SAM?**
>
> We agree with the reviewer that a fully-engineered system should definitely combine the best
> segmentation masks with the best embeddings. If pixel-perfect masks are a requirement, then SAM is definitely the best model to apply, but if representation learning is the goal, models based on SSL offer better features.  In the context of this research, we show that clustering is actually able to retrieve objects competitively and with no supervision. In contrast, SAM is a fully supervised model and requires more than a billion of annotated object masks for training. Our method inherits the advantage of self-supervised learning such as generalizability and low data-collection cost. Furthermore, it is not biased towards a fixed training set and can be scaled without costly human annotation efforts. Similar to the way self-supervised representation learning has gradually replaced supervised representation learning, it is likely that with further research and scaling effort, unsupervised methods such as ours will surpass supervised methods such as SAM.
>
> **Representation evaluations in table 1: since the CLS token is included in the representation set, it is expected to perform similarly in single-object classification tasks (e.g., ImageNet). For multi-object datasets like COCO, centroids should enhance performance by adding more granularity, whereas the CLS token summarizes the entire scene. Therefore, the comparison with the CLS token in Table 1 doesn’t provide much insight.**
>
> We agree with the reviewer that a compositional representation consisting of a global view and multiple object-centric views should, intuitively, should summarize the image better than the CLS token alone. We also agree that this granular representation should affect performance on datasets whose images depict complex scenes in COCO.
>
> These intuitions, however, need to be verified with empirical evidence, as any scientific hypothesis. From the results in Table 1, we confirm that our compositional embeddings lead to similar performance as the CLS token on single-object classification, and improve performance in multi-object datasets compared to using the CLS token alone. The rows corresponding to the CLS token in Table 1 represent a meaningful reference point for all tasks and models considered.
>
> In our opinion, this result is not trivial, though expected, and its inclusion in Table 1 is therefore necessary to provide validation to the expriments. In fact, if the object vectors are not meaningful enough, they can even harm the final representation. In the revised Table 1, we report additional experiments where we concatenate the CLS token with a multi-scale representation from an ensemble of SPOT models. Despite the multi-scale granular representation, this approach fails to match the performance of the CLS token alone.

---

> ### Author Response · Authors · 2024-11-28
> **Response to reviewer srbe (cont.)**
>
> **SPOT is at a disadvantage in table 1. SPOT should include multi-scale representations. A multi-scale version of prior object-centric representations, such as a collection of DINOSAUR slots with S=1,2,4,8 is expected to provide richer information than k-means clustering.**
>
> Following the reviewer’s suggestion, we have trained SPOT with different numbers of slots and used their combined representations in the evaluations in Table 1.  We show the results in the table below. Since the dimension of slot embeddings in SPOT (256) is smaller than the dimension of the CLS token and the average patch embedding from the backbone (768), we simply repeat the slot embeddings 3 times to match the CLS token’s dimension for sake of concatenation. Alternatively, we tried training SPOT models with slot dimension 768, but it does not converge properly, as discussed in [1].
>
> |                         | Tokens      | ImageNet | SUN 397  | Places 205 | CLEVR    | COCO 17  |
> | ----------------------- | ----------- | -------- | -------- | ---------- | -------- | -------- |
> | **DINO ViT-B/16**       | CLS         | **78.2** | **66.9** | 56.0       | 78.2     | 70.6     |
> | + SPOT                  | 7           | 66.8     | 59.7     | 51.1       | 70.8     | 67.6     |
> | + SPOT                  | 8           | 67.6     | 59.6     | 50.8       | 73.7     | 67.7     |
> | + SPOT                  | 16          | 69.2     | 60.3     | 51.5       | 81.5     | 67.7     |
> | + SPOT ensemble         | CLS+1+2+4   | 71.4     | 59.4     | 54.6       | 75.9     | 69.4     |
> | + SPOT ensemble         | CLS+1+2+4+8 | 72.7     | 59.2     | 55.5       | 84.4     | 70.8     |
> | + K-means               | CLS+1+2+4   | 78.1     | 66.0     | 58.8       | 85.6     | 72.3     |
> | + K-means               | CLS+1+2+4+8 | **78.2** | 66.0     | **59.2**   | **89.7** | **72.6** |
> | **DINOv2 ViT-B/14 reg** | CLS         | 83.9     | 77.4     | 64.4       | 74.6     | 77.0     |
> | + K-means               | CLS+1+2+4   | 84.3     | **77.9** | 65.9       | 84.1     | 81.3     |
> | + K-means               | CLS+1+2+4+8 | **84.6** | 77.7     | **66.1**   | **88.2** | **82.3** |
>
> As expected, adding CLS token and using an ensemble of SPOT models generally leads to better performance than single-scale SPOT. However, these results still lag far behind ours. It is also noteworthy that concatenating SPOT object vectors to the CLS token could harm the representation, leading to degradation of performance in multiple datasets. We note that we mainly use SPOT in these multi-scale experiments because the official SPOT codebase is more straightforward to use than the DINOSAUR one, which also does not provide public checkpoints to validate a correct reproduction. We also remark that training a collection of DINOSAUR/SPOT models is time-consuming (at least 24h for training each model) and impractical at inference time.
>
> References:
> - [1] Bridging the Gap to Real-World Object-Centric Learning, Seitzer et al.
>
> **For video action recognition (Figure 3), how does dropping some portion of tokens in the “all-patches” setting affect the tradeoff between token count and performance?**
>
> Thanks for the suggestion. We ran this experiment and compiled the results on video action recognition in Figure 5 in the appendix. Specifically, for each video we drop patches at random to match the number of tokens of the k-means representation. As expected, the performance drops dramatically since the small fraction of remaining patches (8, 16 or 32 out of 256 from each frame) can not represent the movements and interactions of entities in a video.
>
> To complement the suggested experiment, we explored another alternative for reducing the number of tokens per frame without dropping information. Simply explained, we downsample the 2d feature maps of each frame to match a given token budget, e.g., we interpolate the 16x16 grid of patches to 4x4 if the budget is 16 tokens per frame. Once again, given the same budget constraints, the object-centric representation based on k-means tokens outperforms the spatially pooled representation.
>
> We believe that these two alternatives provide convincing evidence that object-centric tokens are powerful and efficient representations for video understanding, and that simply reducing the number of tokens is not sufficient to achieve the optimal trade-off.

---

> ### Author Response · Authors · 2024-11-28
> **Response to reviewer srbe (cont.)**
>
> **DINOv1 vs DINOv2 as a backbone**
>
> In this work, we show that object-centric representations can be achieved with a simple clustering applied to features from powerful general backbones instead of relying on sophisticated fine-tuning procedures. As our main model, we choose DINOv2 since it is the state-of-the-art SSL encoder at the time of writing. For a fair comparison to baselines such as SPOT and DINOSAUR, which are built on top of DINOv1, we also ran the representation learning evaluations on DINOv1 (initial submission, table 1). For completeness, as requested in the review, we also run the unsupervised segmentation evaluations on a DINOv1 backbone and update the results in table 2. For ease of discussion, we report a portion of the updated table below, please consult the revised pdf for results on all six datasets. When using the same DINOv1 backbone, our method still significantly outperforms SPOT and ODIN.
>
> |                           | COCO mBO^i | COCO mBO^c | COCO DetRate^i | VOC mBO^i | VOC mBO^c | VOC DetRate^i |
> | ------------------------- | ---------- | ---------- | -------------- | ---------- | ---------- | ------------ |
> | ODIN (ViT-B/14)           | 49.7       | 54.7       | 48.2           | --         | --         | --           |
> | SPOT (DINOv1 ViT-B/14)    | 46.3       | 51.9       | 46.0           | 56.3       | 59.5       | 64.8         |
> | K-means (DINOv1 ViT-B/14) | 50.2       | 58.9       | 51.1           | 61.3       | 65.3       | 69.8         |
> | K-means (DINOv2 ViT-B/16) | **51.3**   | **61.8**   | **55.0**       | **61.8**   | **66.3**   | **71.0**     |

---

> > ### Comment · Reviewer_srbe · 2024-12-02
> > **Final Score**
> >
> > Thank you for the response and the additional experiments. I have carefully reviewed the authors’ rebuttal as well as the other reviews. As I mentioned in my initial review, this work does not offer any significant novelty or meaningful advancements to the field. Rather, it provides an evaluation of a well-known concept, as acknowledged by the authors in their initial response. Therefore, I am maintaining my original rating of reject.

---

### Official Review · Reviewer_pf52 · 2024-10-31

**Soundness:** 2
**Presentation:** 3
**Contribution:** 2
**Rating:** 5
**Confidence:** 4

**Summary:**

This paper presents a simple and effective clustering-based method for obtaining object-centric feature representations from state-of-the-art ViT models trained with self-supervision. While existing object-centric learning methods are designed with specialized architectures or training techniques, they are limited to specific granularity and small flexibility. The proposed method takes patch features from state-of-the-art ViT models and applies multiple steps of the K-means algorithm to obtain a small set of cluster centroids that represent objects at different levels of granularity. The proposed method shows efficacy at downstream tasks such as image and video classification, with much lower computational cost compared with the baselines.

**Strengths:**

In comparison with learning-based methods, the proposed method offers a set of advantages. 1) It doesn't require any training or fine-tuning to extract object features from the off-the-shelf models. 2) It also renders object features with different levels of granularity on multiple objects/parts with multiple running of the k-means algorithm. 3) The centroids of the extracted features hold semantics that can be applied to downstream tasks.

**Weaknesses:**

1) There is a lack of interpretability and expandability of the "object-centric" features extracted from ViT models. Current ViT models DINO and DINOv2 take image patches as input and are optimized with self-supervised losses. The self-attention layers extract the features at the image patch level, not the object level.  Simply applying K-means clustering on patch features does not guarantee that these features stand for the objects in the images. Wondering about the rationale behind why these features are defined as "object-centric features"?

2) The whole framework of the proposed method relies on the "well-trained features" from ViT models. With some epochs of clustering, these features are used for downstream tasks. Despite its simplicity, I wonder if further steps involving fine-tuning could be taken to update the ViT backbone. This is meaningful as it can help refine the self-supervised ViT features to bring out more semantic information.

3) What is the motivation for choosing video action recognition as one of the experiment setups? To evaluate the semantics quality of object-centric features, it may be a good idea to evaluate on more segmentation tasks on video or image datasets.

**Questions:**

1) Is it possible to evaluate more video/image-based segmentation tasks to show the quality of the semantics of the learned features?

2) What layer of self-attention to extract the patch tokens? Using keys, queries, or values?

3) Time and efficacy analysis: comparison with the baselines?

4) Ablation to the number of running clustering K?

---

> ### Author Response · Authors · 2024-11-28
> **Response to reviewer pf52**
>
> Dear Reviewer `sd7c`, thank you for investing the time in reviewing our manuscript and offering constructive feedback. We have carefully considered your comments and we respond to them below. We welcome further discussion and look forward to your thoughts.
>
> **Clarify patch-level features vs object-centric features: applying K-means clustering on patches does not guarantee that these features stand for objects. Wondering about the rationale behind why these features are defined as "object-centric features"?**
>
> Even though SSL models such as DINOv2 output patch-level tokens, not object-level features, there is convincing evidence that grouping these patch tokens through clustering leads to object-centric features. First, strong performance in semantic segmentation with linear probing on frozen patch features (DINOv2, table 10) demonstrates that patch tokens are representative of the objects covering them, i.e., corresponding tokens of two patches in the same objects are close in the embedding space while tokens of patches in different object classes or background are far away from each other. Second, works in object discovery such as LOST or TokenCut show that object “cutouts” can be obtained by grouping image patches with high-similarity features. This empirical evidence suggests that similar patches in the same cluster should correspond to the same object or object class, and the cluster containing them should correspond to an object or a group of objects.
>
> We agree with the reviewer that even with this insight, there is no guarantee that our object vectors actually correspond to objects. We therefore have dedicated a significant portion of the paper to verify our hypothesis. We summarize our findings below:
>
> 1. K-means centroids focus on distinct objects, and object embeddings preserve semantic information. Consider the retrieval example from Figure 1: for each centroid we visualize the associated image segment, i.e. the object that it supposedly represents, and the closest ImageNet sample retrieved based on similarity in embedding space. It can be seen that masks obtained with our method correspond well to objects in images such as laptops or foods. The accompanied vectors also meaningfully represent the corresponding objects, evidenced by the retrieval results that show food masks retrieve more food and desk masks retrieve more scenes with desks.
>
> 2. K-means cluster assignments are well aligned with human-annotated object masks, as shown by the unsupervised segmentation results in Table 2 and in the qualitative visualizations in Figure 4.
>
> **Suggestion to fine-tune the backbone**
>
> We believe that the reviewer’s suggestion to fine-tune the backbone is a good direction for future exploration. We expect that fine-tuning the whole model on a specific task might lead to performance improvements and more refined features. For example, [1] uses clustering in the loss formulation to encourage feature aggregation. However, the simplicity of our method is a deliberate choice to convey our message: That a method as simple as ours could produce better representation than more involved and optimized techniques shows that previous works overly focus on object mask quality while mostly ignoring the representation quality. Through these results, we invite the object-centric learning community to rethink the current approach to this problem. Our findings also hint at an alternative approach, one which involves first building object-aware general representation learning then extracting objects with simple methods like ours. For more discussion on this point, we refer the reviewer to our general responses to all reviewers.
>
> References:
> - [1] Object discovery and representation networks, Hénaff et al.

---

> ### Author Response · Authors · 2024-11-28
> **Response to reviewer pf52 (cont.)**
>
> **What is the motivation for choosing video action recognition as one of the experiment setups?**
>
> Tacking video inputs using vision transformers is computationally heavy due to the quadratic scaling with respect to the number of patches and video frames.
> In this context, object-centric representations are extremely promising: being able to summarize videos using a structured representation focused around objects, i.e. potentially active entities in a video, is beneficial for training in terms of both compute and memory requirements.
> Thanks to the lightweight runtime of k-means, we were able to evaluate action recognition in videos in several settings (frame resolution, number of frames), and therefore we decided to include this task as evidence of the usefulness of object-centric representations.
>
> **To evaluate the semantic quality of object-centric features, it may be a good idea to evaluate on more segmentation tasks on video or image datasets.**
>
> We have included in the paper a wide range of evaluations to validate our method. Compared to previous works which consider only unsupervised segmentation tasks, we have additionally added benchmarks that probe directly the quality of the representations. Following the reviewer’s suggestion, we extended Table 2 to include unsupervised segmentation evaluations on ClevrTex, MOVI C, MOVI E, and ADE20K. On these datasets, our method produces object masks with higher mBO (instance) and detection rate than those obtained from an expensive ensemble of eight SPOT models, confirming again its effectiveness.
>
> **What layer of self-attention to extract the patch tokens? Using keys, queries, or values?**
>
> In this work, we employ only the patch token features provided by the backbone, without performing additional extraction of key/query/value matrices of the model, in order to keep the approach as simple as possible. Following the reviewer’s suggestion, we have additionally tested key, query and value from the last self-attention as patch features. We added the full results on all datasets in the appendix and report an extract of the table here for simplicity:
>
> |        | COCO mBO^i | COCO mBO^c | COCO DetRate^i | VOC mBO^i | VOC mBO^c | VOC DetRate^i |
> | ------ | ---------- | ---------- | -------------- | --------- | --------- | ------------- |
> | Query  | 33.8       | 39.1       | 22.7           | 43.7      | 46.5      | 35.6          |
> | Key    | 30.7       | 36.0       | 19.9           | 41.8      | 45.1      | 34.5          |
> | Value  | 26.5       | 32.1       | 16.2           | 37.6      | 41.8      | 29.4          |
> | Output | **51.3**   | **61.8**   | **55.0**       | **61.8**  | **66.3**  | **71.0**      |
>
> In the setting considered in this work, the patch features prove to be the best choice for clustering-based unsupervised segmentation, which we attribute to the clustering-like effect of the IBOT loss applied to patch features during DINOv2 pre-training
>
> **Time and efficacy analysis: comparison with the baselines?**
>
> Compared to the baselines, our approach requires no training nor fine-tuning, and, at runtime, the proposed multi-scale k-means runs with little overhead (clustering time is negligible compared to the forward pass of a ViT) and yields objects at all desired granularities. In contrast, SPOT requires at least 24h of training to produce one model capable of predicting objects at a single granularity as defined by its hyperparameters (e.g. number of slots). Switching to a different granularity requires training another model and running inference with several models in parallel, which is impractical for real-world scenarios.
>
> **Ablation to the number of running clustering K?**
>
> The results for image classification in Table 1 and for video action recognition in Figure 3 already present the results of our method with 8 vs 16 clusters. Furthermore, Figure 2 offres a more comprehensive comparison using 4, 8, 16 and 32 clusters on several datasets. We have found that adding more object vectors generally leads to better performance, especially on datasets of complex scenes such as COCO or SUN397, or on video action recognition tasks where large spatial and temporal resolutions are crucial. In the updated pdf, we extended the range of k-means clusters for the video evaluation, which now uses 4, 8, 16, 32, or 64 clusters (Figure 5 in the appendix).

---

### Official Review · Reviewer_ziUG · 2024-11-04

**Soundness:** 3
**Presentation:** 3
**Contribution:** 2
**Rating:** 3
**Confidence:** 4

**Summary:**

This paper reveals that directly applying the classic K-Means clustering algorithm to DINOv2 features can yield meaningful object-centric representations. To enhance flexibility, the authors propose a multi-round K-Means clustering pipeline that utilizes varying K values and hierarchical resolutions. Experiments demonstrate the characteristics of the object-centric embeddings captured through K-Means clustering and compare these results to those obtained from learning-based methods.

**Strengths:**

- The motivation is clear, and the method is straightforward to understand.
- Extensive experiments and thorough analysis are conducted to explore the characteristics of embeddings obtained by applying K-Means on DINO V2 features, along with comparisons to those captured by learning-based methods.

**Weaknesses:**

- The main weakness of this work is the lack of novelty. DINOv2 features are already well-established for their ability to be clustered to represent objects, as evidenced not only by the original figures in DINO but also by numerous studies (both learning-based and non-parametric) in the field of unsupervised segmentation that have explored this capability. Consequently, it is expected that a simple K-Means clustering algorithm can be applied directly to DINOv2 features to yield meaningful object-centric representations.
- K-Means clustering also offers limited flexibility, as the choice of the number of clusters K and the design of the hierarchical structure to achieve optimal performance are hyperparameters that vary per dataset. Furthermore, whether the origin information contained in the resulting object embeddings is closely tied to the specific downstream tasks being addressed, making it difficult to assert that maintaining full semantic information is a definitive advantage.

**Questions:**

Please refer to the weaknesses.

---

> ### Author Response · Authors · 2024-11-28
> **Response to reviewer ziUG**
>
> Dear reviewer `ziUG`, we appreciate the time you took to review our manuscript and provide valuable feedback. In this response, we aim to address your main concerns and look forward to continuing the discussion.
>
> **Main weakness is lack of novelty. It is expected that a simple K-Means clustering algorithm can be applied directly to DINOv2 features to yield meaningful object-centric representations.**
>
> We agree that it’s not surprising that applying k-means on DINOv2 embeddings would result in meaningful representations that perform well on downstream tasks, since k-means centroids live in the embedding space of DINOv2. What is surprising, and at the core of the message of the paper, is that other approaches for object-centric learning produce worse object embeddings than k-means. The novelty of the paper is to provide more holistic evaluations, in the hope to shift the focus back where it matters: from maximising dataset-specific unsupervised segmentation metrics, to actually evaluating the learned representations. It may well be that additional layers, training stages and custom losses are not necessary for object-centric learning.
>
> For an extended discussion regarding the paper’s novelty and contributions, we refer the reviewer to our general response in the main comment.
>
> **K-Means clustering also offers limited flexibility, as the choice of the number of clusters K and the design of the hierarchical structure to achieve optimal performance are hyperparameters that vary per dataset.**
>
> We respectfully disagree with this comment. Contrary to the object-centric learning literature, we do not select hyperparameters for each dataset. For all evaluations focused on representations (section 4.1), we use the same method identically for all datasets: multiple k-means clustering with a number of clusters that follow powers of two. The same approach is used for the unsupervised segmentation evaluations in section 4.2, table 2.
>
> The only setting in which we fix the number of clusters on a dataset-by-dataset basis is the unsupervised segmentation in table 3. This is only done to facilitate comparison with previous works, since it has become common practice in the object-centric literature to fix the number of slots for each dataset. In the manuscript (initial submission, L441-444), we have also included our critique of this evaluation protocol, which encourages dataset-specific hyperparameter tuning rather than generalization.
>
> On the topic of hyperparameters, we invite the reviewer to compare k-means clustering with DINOSAUR and SPOT, which are considered state-of-the-art methods in the object-centric learning literature:
> - k-means: simple implementation, fast runtime, few parameters to tweak (number of clusters, initialization), it takes no more than a few minutes to try many hyperparameter settings
> - DINOSAUR/SPOT: requires a custom slot attention layer, many hyperparameters (number of slots, train dataset, learning rate, number of iteration, slot dimension, etc.)  it takes around 24h to get a result for a single hyperparameter setting.
>
> **The origin information contained in the resulting object embeddings is closely tied to the specific downstream tasks being addressed, making it difficult to assert that maintaining full semantic information is a definitive advantage.**
>
> The reviewer raises a reasonable concern which applies more in general to self-supervised learning as a whole: how can we be sure that features learned with a general pretext task will be useful for downstream tasks? Are all the features useful or are some of them redundant?
> The SSL literature answers this concern by evaluating as many downstream tasks as possible, hoping to reduce uncertainty about the quality of the learned representations by accumulating evidence over a large and diverse set of experiments.
>
> In the same spirit of SSL, in section 4.1, we tried to provide an extensive suite of evaluation tasks, from single-object image recognition to temporal action recognition in videos. For all downstream tasks considered in this section, the information captured by the k-means object vectors has proven to be superior to the one obtained from methods based on slot-attention, which aim to compress and reduce information from the backbone. This result suggests that the full semantic information is useful, and it's better to preserve it.
>
> At the same time, we acknowledge that there are other tasks and data domains that are not covered by our evaluations, such as robotics, reinforcement learning, or visual question answering. We expect that better SSL backbones will enable object-centric features to generalize to those tasks as well. Further verifying this claim with a more exhaustive set of experiments would be an exciting avenue for research on its own.

---

### Official Review · Reviewer_sd7c · 2024-11-04

**Soundness:** 1
**Presentation:** 3
**Contribution:** 2
**Rating:** 3
**Confidence:** 4

**Summary:**

This paper presents a simple method that clusters feature from the representation from existing backbones, i.e., K-means clustering based on the extracted feature. The K is a fixed value. Some experiments are conducted to claim that this simple clustering method performs better than many other methods requiring additional training.

**Strengths:**

1. The presentation is good. The story is very clear to understand.
2. The method is simple, and simplicity means easy to be applied to downstream fields.

**Weaknesses:**

1. Claiming a kind of generally learned representation is better needs numerous diverse experiments to fully evaluate the method from different perspectives.  The experiments, which include only single-label classification and unsupervised segmentation, are insufficient. I think this work just selects a few experimental settings supporting the desired conclusion.
2. Clustering based on the representation produced by existing backbones is a commonly used baseline in fields like few-shot and unspervised learning. There is no new information in this paper.

**Questions:**

I recommend the authors delving deeper into the studied problem to reveal some insightful observations.

---

> ### Author Response · Authors · 2024-11-28
> **Response to reviewer sd7c**
>
> Dear reviewer `sd7c`, thanks for investing time in reading our manuscript and giving feedback. In the following paragraphs, we try to address the main concerns and we look forward to engaging in discussion further.
>
> **Insufficient experiments to claim generality, only single-label classification and unsupervised segmentation.**
>
> We respectfully disagree with the reviewer’s comment. Not only does this work include all unsupervised segmentation evaluations that are well-established in the object-centric learning literature, but it expands the evaluation suite to cover the often-overlooked aspect of representation learning.
> As explained in the general response to all reviewers, this work re-establishes the two-sided nature of object-centric learning: representation learning and object localization. To this end, we have actually added several evaluations that are missing in previous object-centric learning works. These include single-label classification for object recognition (ImageNet) and scene recognition (SUN397, Places205), multi-label object recognition (CLEVR, COCO 2017), and video action classification (Kinetics 400 and SomethingSomething v2). Following other works, we have also provided experimental results on unsupervised segmentation benchmarks such as VOC, COCO, MOVI-C and MOVI-E under two complementary evaluation protocols.
>
>
> **Clustering based on the representation produced by existing backbones is a commonly used baseline in fields like few-shot and unsupervised learning. There is no new information in this paper.**
>
> K-means clustering is commonly portrayed as a weak baseline in the object-centric literature, which mainly evaluates unsupervised segmentation tasks and ignores the quality of the learned representation. The new information in this paper is that a) k-means actually works quite well as an unsupervised segmentation method, provided that the evaluation protocol allows for a high-recall scenario, and b) the object embeddings produced by k-means are superior to the ones produced by more complicated object-centric learning approaches. These results suggest rethinking the current approach to object-centric learning. On one hand, future research should pay more attention to the evaluation of the learned representation. On the other hand, our findings hint at a shift to first building object-aware general representation learning then extracting objects with simple methods like ours. For an extended discussion regarding the paper’s novelty and contributions, we refer the reviewer to our general response in the main comment.

---

### Author Response · Authors · 2024-11-28
**Common response to the reviewers**

We thank the reviewers for their constructive comments and actionable feedback. We address the common concern about novelty and technical contribution below, and provide individual responses to other reviewers' feedback in separate comments.

## Novelty and contributions
In this work, we do not aim to introduce a novel method for object-centric learning, e.g. a new architecture or loss formulation. Instead, our focus is to bring an important perspective to the object-centric learning literature and re-consider a simple approach based on clustering as a strong baseline.

Object-centric learning, at its core, seeks to learn a structured representation of visual scenes by decomposing images into segments (objects, parts, or groups of objects) and attaching to each segment an embedding that represents it. The goal is therefore two-fold: the decomposition into segments and their representation.

In recent years, the most prominent works in the object-centric learning literature [1-5] have been focusing on the unsupervised segmentation aspect, i.e. evaluating whether the predicted segments effectively match objects in annotated datasets. Maximising segmentation metrics such as FG-ARI and mIoU has led to the development of increasingly sophisticated techniques, involving multi-stage training and custom layers. On the other hand, the quality and usefulness of the learned representation has been greatly overlooked. Only a few works attempt to probe the learned representation with ad-hoc prediction tasks, but no established benchmark and common metric to track exist.

In our work, we re-establish this two-sided nature of object-centric learning, and treat the evaluation of learned object embeddings as a first-class citizen. We show that the embeddings learned by all recent object-centric methods produce underwhelming results on all tasks considered, often being surpassed by a global representation such as a single CLS token.
Then, we consider the embeddings produced by k-means clustering, an algorithm that is often presented as a weak baseline in object-centric learning papers, and demonstrate that if properly applied it can outperform all recent object-centric learning approaches, while being at the same time simpler, faster and more efficient.

Therefore, the main contribution of this paper is the novel perspective on what really matters in object-centric learning, and what is missing in contemporary works. Our results suggest rethinking the current approach to object-centric learning. On one hand, future research should pay more attention to the evaluation of the learned representation. On the other hand, our findings hint at a shift to first building object-aware general representation learning then extracting objects with simple methods like ours.

## Updated pdf revision following the rebuttal
We integrated the suggestions from the reviewers and updated the manuscript according to their feedback. We uploaded two revisions that are identical in content, but the first revision highlights all modifications in orange color for ease of review. We summarize the changes below:
- Added a sentence in the abstract and in the introduction to better clarify novelty and contributions (reviewers `sd7c` and `ziUG`)
- Added references mentioned in the discussion (reviewer `srbe`)
- Additional experiments on representation quality in table 1 to evaluate the multi-scale representation of an ensemble of SPOT models (reviewer `srbe`)
- Extended and clarified the comparison between DINOv1 and DINOv2 in the representation learning evaluation in table 1 and unsupervised segmentation evaluation in table 2 (reviewers `pf52` and `srbe`)
- Additional datasets for the unsupervised segmentation experiments in table 2: ClevrTex, ADE20K (reviewer `pf52`)
- Ablation study in the appendix: video action recognition using a random selection of patches as the video representation (reviewer `srbe`)
- Backbone comparison for SPOT in the appendix: training SPOT on top of DINOv1 vs DINOv2 (reviewer `srbe`)
- Ablation study in the appendix: use query/key/values from the last self-attention layer instead of output patch features (reviewer `pf52`)

## Individual comments and discussion
We address all the points raised by the reviewers in individual comments under each review, providing clarifications, context, and additional experiments as needed. We encourage the reviewers to follow-up on our answers and look forward to the ensuing discussion.

## References
- [1] Object-Centric Learning with Slot Attention, Locatello et al.
- [2] Bridging the Gap to Real-World Object-Centric Learning, Seitzer et al.
- [3] SPOT: Self-Training with Patch-Order Permutation for Object-Centric Learning with Autoregressive Transformers, Kakogeorgiou et al.
- [4] Object discovery and representation networks, Hénaff et al.
- [4] Object-Centric Slot Diffusion, Jiang et al.
- [5] Illiterate DALL-E Learns to Compose, Singh et al.

---

### Meta-Review · Area_Chair_pVsV · 2024-12-22

**Metareview:**

This paper presents a simple method that clusters features from existing backbones for obtaining object-centric feature representations. The manuscript was reviewed by four experts in the field. The recommendations are (3 x "3: reject, not good enough", "5: marginally below the acceptance threshold"). The reviewers raised many concerns regarding the paper, e.g., limited technical novelty, unclear motivation, unconvincing experimental evaluations, inadequate literature reviews, etc. Considering the reviewers' concerns, we regret that the paper cannot be recommended for acceptance at this time. The authors are encouraged to consider the reviewers' comments when revising the paper for submission elsewhere.

**Additional Comments On Reviewer Discussion:**

Reviewers mainly hold concerns regarding limited technical novelty (Reviewers sd7c, ziUG), unclear motivation (Reviewers pf52, srbe), unconvincing experimental evaluations (Reviewers sd7c, ziUG, pf52, srbe), inadequate literature reviews (Reviewer srbe). The authors' rebuttal could not fully address the above-listed concerns.

---

### Decision · Program_Chairs · 2025-01-22

Reject